# Association of Chronic Kidney Disease with Prior Tinnitus: A Case–Control Study

**DOI:** 10.3390/jcm11247524

**Published:** 2022-12-19

**Authors:** Shih-Han Hung, Sudha Xirasagar, Yen-Fu Cheng, Nai-Wen Kuo, Herng-Ching Lin

**Affiliations:** 1Department of Otolaryngology, School of Medicine, Taipei Medical University, Taipei 110, Taiwan; 2Department of Otolaryngology, Wan Fang Hospital, Taipei Medical University, Taipei 110, Taiwan; 3International Master/Ph.D. Program in Medicine, College of Medicine, Taipei Medical University, Taipei 110, Taiwan; 4Department of Health Services Policy and Management, Arnold School of Public Health, University of South Carolina, Columbia, SC 29208, USA; 5Department of Medical Research, Taipei Veterans General Hospital, Taipei 112, Taiwan; 6Department of Otolaryngology-Head and Neck Surgery, Taipei Veterans General Hospital, Taipei 112, Taiwan; 7Department of Otolaryngology-Head and Neck Surgery, School of Medicine, National Yang Ming Chiao Tung University, Taipei 112, Taiwan; 8Institute of Brain Science, National Yang Ming Chiao Tung University, Taipei 112, Taiwan; 9Research Center of Sleep Medicine, College of Medicine, Taipei Medical University, Taipei 110, Taiwan; 10School of Health Care Administration, College of Management, Taipei Medical University, Taipei 110, Taiwan; 11Sleep Research Center, Taipei Medical University Hospital, Taipei 110, Taiwan

**Keywords:** tinnitus, chronic kidney disease, epidemiology

## Abstract

This population-based, case–control study aims to explore the relationship between prior tinnitus and the occurrence of chronic kidney disease (CKD) using a nationwide, population-based cohort study. We used data from the Taiwan National Health Insurance Research Database to explore the association of CKD with tinnitus. We identified 15,314 patients aged ≥40 years old with a first-time diagnosis of CKD as the cases. We used propensity-scored matching to select 45,942 controls (1:3 ratio). We performed multivariate logistic regression to estimate the odds ratio (OR) of a prior tinnitus diagnosis among the CKD group vs. the control group. Analysis showed that 770 (1.26%) out of the 61,256 sampled patients had previously diagnosed tinnitus. Chi-square testing revealed a significant difference in the rate of previously diagnosed tinnitus between cases and controls (3.86% vs. 0.93%, *p* < 0.001). Univariate logistic regression analysis showed an OR of prior tinnitus for cases of 10.249 (95% confidence interval (CI): (8.662~12.126)) relative to controls. In adjusted analysis, cases were more likely than controls to have a prior diagnosis of tinnitus (OR = 10.970, 95% CI = 9.255~13.004, *p* < 0.001) after adjusting for age, sex, monthly income, geographic location, urbanization level, hypertension, diabetes, coronary heart disease, hyperlipidemia, obesity, and autoimmune disease. Our study shows that CKD patients have a higher likelihood of having suffered from tinnitus before CKD was diagnosed, but we have no data suggesting that tinnitus is a predictor of subsequent CKD. Patients diagnosed with tinnitus may benefit from proactive measures to prevent CKD and detect it early through lifestyle modifications and regular renal function examinations, regardless of CKD-related symptoms.

## 1. Introduction

Chronic kidney disease (CKD) affects 8% to 16% of the world’s population and is the 16th leading cause of life-years lost worldwide [1]. It has been reported that Taiwan has the highest incidence and prevalence rates of end-stage renal disease (ESRD) in the world [2]. Diabetes and hypertension are the leading causes of CKD in all high-income and middle-income countries, and in many low-income countries. With increasing populations affected by diabetes and hypertension globally, CKD prevalence is expected to substantially increase in the near future, with serious implications for quality of life, healthcare costs, and premature mortality [3].

Given the systemwide microangiopathy impacts of diabetes and hypertension, and its specific role in the pathogenesis of CKD, as expected, organs susceptible to microangiopathy, such as the retina, have shown comorbid conditions (e.g., retinopathy) with CKD [4,5,6]. Similarly, certain inner ear diseases are also related to microangiopathies [7,8]. The relationship between CKD and inner ear diseases has been explored previously. For decades, an elevated chance of hearing loss and related disorders has been observed among patients with chronic kidney failure [9,10,11,12]. In animal studies, prescription drugs used to support kidney function are shown to be effective in protecting hair cells of the Corti organ against aminoglycoside antibiotic ototoxicity, and are shown to protect hearing function [13]. Salicylate is also reported to protect both hearing and kidney function from cisplatin toxicity, independent of its oncolytic action [14]. These studies indicate a shared susceptibility between the inner ear and the kidney via injury pathways common to both organs.

While the link between hearing impairment and CKD has been considerably studied, little is known regarding tinnitus and CKD. Tinnitus, the subjective experience of noise without an extraneous source of the experienced noise, is often generated by the same underlying conditions that cause hearing loss, such as age, ear injury, or problems with the circulatory system. While multiple pathophysiologies have been identified, including inner ear pathology, auditory nerve synchronization, central nervous system anomalies, and limbic and autonomous nervous system problems, tinnitus, with a prevalence range from 7.1% to 14.6%, remains poorly understood in terms of its underlying mechanisms responsible for the development of this abnormal sensory state [15]. Patients with CKD were shown to have increased rates of tinnitus in one nationwide cohort study [16]. However, there would be significant clinical value in findings that can provide guidance to prevent the subsequent development of CKD with evidence that goes beyond cross sectional associations. This may provide a survival advantage if tinnitus, which often long precedes hearing loss or is unrelated to hearing loss in some cases, can be used to predict future disease, including CKD. It is possible that the presence of tinnitus before the development of related comorbidities may define a phenotype of higher vulnerability to microangiopathies, permitting the use of tinnitus as a marker for future related co-morbidities, including CKD.

This study aims to explore the relationship between prior tinnitus and the occurrence of CKD using nationwide, population-based cohort data.

## 2. Materials and Methods

### 2.1. Database

This study used data from the Taiwan National Health Insurance (NHI) Research Database (NHIRD) to explore the association of subsequent CKD with tinnitus. In 2021, ~99.99% of Taiwan’s citizens (23,876,603 individuals) were enrolled in the NHI program. The NHIRD includes registries of beneficiaries, ambulatory care claims, inpatient claims, prescriptions dispensed at pharmacies, medical facilities, and board-certified specialists. Numerous researchers from Taiwan have used the NHIRD to carry out longitudinal studies of diseases and care effectiveness using claims data from follow-up medical services. Currently, only Taiwanese nationals are permitted direct access to these data.

The study was approved by the institutional review board of Taipei Medical University (TMU-JIRB N202203074), and is compliant with the Declaration of Helsinki. Because we used de-identified administrative data provided for research, informed consent was waived.

### 2.2. Identification of Cases and Controls

We identified all patients aged ≥40 years old with a first-time diagnosis of CKD (ICD-9-CM code 585, 403.01, 403.11, 403.91, 404.02, 404.03, 404.12, 404.13, 404.92, 404.93 or ICD-10-CM codes N18.5, N18.6, I12.0, I13.11, I13.2) during an ambulatory care visit between January 2015 and December 2016, 16,186 patients. We assigned the first date of the CKD diagnosis as the index date. In light of potential concerns regarding the reliability of diagnosis coding, we refined the inclusion criteria to include only patients with at least two different claims showing a CKD diagnosis by board-certified nephrologists. As a result, we included 15,314 patients with CKD.

We selected controls from the remaining NHI enrollees from the registry of beneficiaries using a controls to cases match ratio of 3:1 (*n* = 45,942). We first excluded all enrollees who had ever received a diagnosis of CKD before the age of 40 years old. For the remaining, propensity-scored matching was performed to identify controls, using demographics (age, sex, monthly income category, geographic location, and urbanization level of the patient’s residence), and medical comorbidities including hyperlipidemia, diabetes, coronary heart disease, hypertension, obesity, and autoimmune disease as matching variables. We included these variables in a multivariable logistic regression model as predictors to calculate the propensity score for CKD for each enrollee. Because exact score-matched controls may not be available for every case, we used the alternative method of nearest neighbor within calipers to match controls (a priori value for the calipers set at +/−0.01). We matched controls to a given CKD patient if they had utilized any ambulatory care service in the index year of the CKD case. The date of the control patient’s first ambulatory care visit during the index year of their matched case was assigned as their index date. Ultimately, the final study sample had 61,256 sampled patients, 15,314 cases, and 45,942 controls.

### 2.3. Ascertainment of Exposure

Our objective was to estimate the odds of the exposure of interest, a previous diagnosis of tinnitus (ICD-9-CM code 388.3 or ICD-10-CM codes H93.1, H93.11, H93.12, H93.13 or H93.19), for cases relative to controls. To improve the veracity of the tinnitus diagnoses, we defined a patient as having tinnitus only if they had at least two claims showing a diagnosis of tinnitus within the 3-year period prior to their index date.

### 2.4. Statistical Analysis

We used the SAS system (SAS for Windows, V, 8.2, SAS Institute, Cary, NC, USA) for statistical analyses. Chi-square and t-tests were carried out to study differences in demographic characteristics and medical comorbidities between cases and controls. We used multiple logistic regression analysis to examine the association of CKD with prior tinnitus after adjusting for age, sex, monthly income category, geographic location and urbanization level of the patient’s residence, hyperlipidemia, diabetes, coronary heart disease, hypertension, obesity, and autoimmune disease. Notably, despite using propensity score matching to select controls, the large sample sizes may produce statistically significant *p*-values despite negligible differences in magnitude or counts. Effect sizes were studied to test the differences based on Cohen’s d, Cohen’s h, or Cohen’s φ. We used *p* ≤ 0.05 to assess statistical significance.

## 3. Results

Table 1 presents the sociodemographic characteristics and medical comorbidities among the sample patients. Mean ages for cases and controls were 58.6 ± 11.1 years and 59.2 ± 10.6 years, respectively (*p* < 0.001, Cohen’s d = 0.055). Regarding sociodemographic characteristics, there were significant differences between cases and controls for sex (*p* < 0.001, Cohen’s h = 0.054), monthly income (*p* < 0.001, Cohen’s φ = 0.025), geographic location (*p* < 0.001, Cohen’s φ = 0.027), and urbanization level (*p* < 0.001, Cohen’s φ = 0.019). Regarding medical co-morbidities, there were significant differences between the cases and controls for presence of diabetes (14.9% vs. 9.5%, *p* < 0.001, Cohen’s h = 0.015), hyperlipidemia (20.3% vs. 12.6%, *p* < 0.001, Cohen’s h = 0.021), hypertension (24.5% vs. 19.7%, *p* < 0.001, Cohen’s h = 0.017), coronary heart disease (5.4% vs. 4.4%, *p* < 0.001, Cohen’s h = 0.046), and obesity (0.4% vs. 0.3%, *p* = 0.015, Cohen’s h = 0.017). Although there were significant differences, all effect sizes were small (≤0.2), and were considered too small to be of practical significance.

Table 2 presents the prevalence of prior tinnitus among cases and controls (Table 2), showing that 770 (1.26% of the total sample) had previously diagnosed tinnitus, 3.86% among cases and 0.93% among controls (*p* < 0.001). We found that of patients with CKD and with tinnitus, the duration between the first diagnosis of CKD and the first diagnosis of tinnitus was 803 ± 547 days. Univariate logistic regression showed an odds ratio (OR) of prior tinnitus of 10.25 (95% confidence interval [CI]: (8.66~12.13)) for cases relative to controls.

Table 3 presents the covariate-adjusted odds of prior tinnitus for CKD patients relative to controls, OR = 10.970, 95% CI = 9.255~13.004 (*p* < 0.001), after adjusting for age, sex, monthly income, geographic location, urbanization level, hypertension, diabetes, coronary heart disease, hyperlipidemia, obesity, and autoimmune disease.

## 4. Discussion

Our study found that patients with a prior presentation of tinnitus were 10-fold more likely to develop CKD, supporting that the inner ear may share with other organ systems the same susceptibility to microvascular injuries that may be caused by various disease pathologies. The findings also support tinnitus as a marker to predict possible future kidney disease. The unique contribution of this study is that while other cohort studies reported an association of CKD with inner-ear problems, our study may be the first to identify a common symptom, tinnitus, as a marker for a higher probability of developing renal disease in the future. This time lag between tinnitus and CKD may enable CKD prevention/precaution measures in susceptible subgroups (e.g., diabetes, hypertension patients) before the life-threatening condition of CKD becomes established.

Globally in 2017, CKD caused an estimated 1.2 million deaths (95% uncertainty interval [UI] 1.2 to 1.3). Across all age groups, global CKD mortality increased by 41.5% (95% UI 35.2 to 46.5) between 1990 and 2017 [17]. The disease burden and cost burden of CKD is enormous enough to have triggered the use of very substantial resources to prevent this disease. Diabetes and hypertension are the leading causes of death in all developed countries. In contrast, CKD arising out of glomerulonephritis (usually occurring in childhood) and unknown causes are more common in Asia and sub-Saharan African countries. The contrast reflects the general shift in the burden of disease from infectious etiology toward chronic lifestyle-related etiology and aging as a result of lower birth rates and increased life expectancy in developed countries [18]. In addition to these metabolic diseases, in Asia, CKD may arise from the use of certain toxic substances and herbs as part of traditional medicine practices, such as herbs containing nephrotoxin and aristolochic acid (present in some commonly used traditional herbal products) [19]. Except for nephropathies of these origins, uncommon CKD of unknown etiology, and mesoamerican nephropathy, it would appear that most other CKDs may be mitigated by advance measures to protect the kidneys from dysfunction. Having advanced knowledge of marker diagnosis or predictor diagnosis may enable persons at risk to adapt their lifestyle and chronic disease control efforts to actively prevent the progression of their underlying conditions. Our study may be significant from this perspective.

Explorations of the relationship between the inner ear and the kidney were triggered by observations of co-occurring hearing loss and kidney disease, such as in Alport’s syndrome, kidney transplantation patients, and dialysis patients [20]. The phenomenon is supported by the fact that some pharmacologic agents, mostly aminoglycosides that act on the transport mechanisms of the renal tubular epithelium, often also cause inner ear disturbances including tinnitus and hearing loss. In 1980, Fee et al. analyzed one hundred and thirty-eight patient courses of tobramycin and gentamicin and reported that ototoxicity was independent of nephrotoxicity, suggesting that inner ear damage is not necessarily a consequence of renal failure [21]. It has been proposed in many studies that aminoglycoside toxicity involves its interaction with specific cell membrane lipids, polyphosphoinositides, which are in much higher concentration in the kidney, neural tissue, and the inner ear than in other tissues [22]. Large-scale population-based observational studies have also supported this link. Lin et al. reported that sudden sensorineural hearing loss was 1.57 times higher in the CKD-affected group than in the CKD-free group in a nationwide population-based study [23]. More recently, Kang et al. studied 16,554 participants and reported that male and female subjects with CKD had higher hearing thresholds than the subjects without these disorders, and CKD was associated with increased hearing thresholds in both men and women [24]. The report by Seo et al. demonstrated a dose-dependent relationship—the odds of hearing impairment were higher among the group with a lower estimated glomerular filtration rate than with a normal estimated glomerular filtration rate (EGFR), and individuals with CKD were more likely to also have a hearing impairment [25]. A similar result was also demonstrated in Chinese populations, showing EGFR and serum creatinine as effective predictors of hearing loss [26].

However, despite evidence of these relationships, it has been a challenge to apply these findings in clinical practice to reduce CKD risk, as there has been no evidence of advanced marker conditions that could be used to flag at-risk individuals for advanced preventive measures. Few studies have explored predictive factors for CKD. Borisagar et al. recently reported on a neural network algorithm for CKD prediction, concluding that CKD detection based on a neural network algorithm is highly accurate, both as an alternative diagnosis path for doctors, and for the lay population to assess the probability of having CKD [27]. In the study, while 24 attributes including age, blood pressure, blood urea, and diabetes were used as the neural network input, hearing/inner ear symptoms were not among them. Hearing-related predictors were also not included in many subsequent machine learning studies to predict CKD [28,29]. Therefore, our finding that prior tinnitus is associated with the subsequent development of CKD is particularly interesting. This is because not only can this unmistakable common clinical symptom, tinnitus, be used as a predictor, but also because this oft-ignored symptom occurs before the diagnosis of CKD. One study examined patients with a traditional Chinese medicine diagnosis of “subjective tinnitus (ST) having kidney deficiency pattern” by analyzing serum metabolic profiles. The authors reported that metabolic pathogenesis in ST/KDP subjects was characterized by upregulated glutamate, serotonin, orotic acid, and 8-oxoguanine, as well as downregulated taurine, and additionally, perturbations of calcium signaling, GABA receptor signaling, purine and pyrimidine biosynthesis, taurine biosynthesis, and serotonin receptor signaling [30].

Our study has several limitations. First, being based on health insurance claims data, there remains the possibility of surveillance bias. Patients with tinnitus may have more exposure to healthcare services, resulting in referral to general internal medicine care or a general health check-up, and resulting in more diagnosis of CKD among the tinnitus-affected population. However, this bias is likely to be limited as, currently, there are no guidelines suggesting that patients with tinnitus should be evaluated for renal function. Second, the medication or substance use history of the study patients is unknown. Some patients with tinnitus may seek traditional Chinese herbal medications for their tinnitus problems, and as reported in some literature, this may be the actual cause of subsequent CKD [31].

Additionally, given that the study is based on health insurance claims data, we have no data on the hearing status of subjects at the time of the tinnitus diagnosis. Tinnitus may result from hearing loss, and this hearing loss may be due to subclinical kidney disease.

Another limitation is that the etiology of CKD is multifactorial, and it is not possible to account for important factors, notably, the use of herbal medicine or nephrotoxic medicines. We could only include covariates with documented relationships with CKD and common medical comorbidities. Some underlying predisposing factors of CKD may be independent to prior tinnitus and are not documented to date or accounted for in this study.

Finally, given that the study is based on claims data, the severity of CKD and its treatment status cannot be addressed. Therefore, we cannot study dose response, or whether patients with severe or prolonged prior tinnitus have a greater risk of CKD than those with less severe cases.

Another point that is important to be stressed is that, in our study, we define tinnitus as patients with a diagnosis of tinnitus and at least two claims showing tinnitus within 3 years, and thus might be regarded as quite severe tinnitus. This defined tinnitus might be quite different from patients who ever experienced a ringing sound in their ears, and could be a problem if questions about tinnitus are ever used as screening questions. Without higher selection criteria, a high prevalence of tinnitus is expected in the population, and as CKD does not cause the majority of tinnitus, to increase the applicability, more data about the degree of tinnitus and hearing loss in these subjects might be necessary for the phenomenon to serve as a proper prognostic tool.

Despite the limitations of claims-based data, the high number of cases and controls studied reinforces the validity of the finding and is a strength of the study. There may be some implications of our study beyond the suggested preventive approach. One implication is to explore a possible phenotype of the co-existing vulnerability of the inner ear and renal system in humans. Genetics has been little explored in the context of tinnitus, with a few potential genes screened, including associated cardiovascular genes, neurotropic factors of BDNF and GDNF, potassium recycling pathway genes, GABA_B_ receptor subunits, and serotonin receptor/transporters [32,33]. Among these, the neurotrophic factor GDNF is associated with kidney development, and BDNF is essential for glomerular development, morphology, and function [34,35]. While these studies provide some pointers, further study is needed to explore the potentially shared pathophysiology and mechanisms of the observed relationship between hearing and kidney function. In addition, importantly, longitudinal studies may help establish whether precautionary measures against kidney disease initiated at the onset of tinnitus would prevent CKD, compared to those without such measures. Finally, survey research of patients with tinnitus and matched controls should be undertaken to understand the prevalence of traditional medicine use by these patients, in addition to screening for use of the known reno-toxic agents among traditional medicine users.

Nevertheless, the high odds of CKD observed in our study suggest the need to consider exploratory attempts, including surveillance for CKD and lifestyle precautionary measures among patients diagnosed with tinnitus, and among persons with existing CKD, as measures to halt disease progression.

In conclusion, our study finds that patients with prior tinnitus are associated with a higher risk of a subsequent diagnosis of CKD. These patients may benefit from specific measures to prevent or detect CKD early through lifestyle modifications and regular renal function examinations, regardless of renal function-related symptoms.

## Figures and Tables

**Table 1 jcm-11-07524-t001:** Demographic characteristics of chronic kidney disease (CKD) cases and controls in Taiwan (*n* = 61,256).

Variable	Patients with CKD (*n* = 15,314)		Controls (*n* = 45,942)		*p* Value
	Total No.	%	Total No.	%	
Age, mean (SD)	58.6 (11.1)		59.2 (10.6)		<0.001
Males	9830	64.2%	28,288	61.6%	<0.001
Monthly Income					<0.001
<TWD 1~15,841	5051	33.0%	13,392	29.1%	
TWD 15,841~25,000	4813	31.4%	16,137	35.1%	
≥TWD 25,001	5450	35.6%	16,421	35.7%	
Geographic region					<0.001
Northern	7296	47.6%	20,726	45.1%	
Central	3438	22.5%	11,325	24.7%	
Southern	4160	27.2%	12,784	27.8%	
Eastern	420	2.7%	1115	2.4%	
Urbanization level					<0.001
1 (most urbanized)	4249	27.8%	12,111	26.4%	
2	4519	29.5%	13,517	29.4%	
3	2676	17.5%	7927	17.3%	
4	1966	12.8%	6364	13.8%	
5 (least urbanized)	1904	12.4%	6031	13.1%	
Diabetes	2278	14.9%	4376	9.5%	<0.001
Hypertension	3756	24.5%	9047	19.7%	<0.001
Coronary heart disease	833	5.4%	2031	4.4%	<0.001
Hyperlipidemia	3106	20.3%	5809	12.6%	<0.001
Obesity	66	0.4%	138	0.3%	0.015
Autoimmune disease	254	1.7%	855	1.9	0.104

**Table 2 jcm-11-07524-t002:** Prevalence of prior tinnitus, crude odds ratios (ORs) for tinnitus, and 95% confidence intervals (CIs) among cases with CKD vs. controls.

Presence of Prior Tinnitus	Total (*n* = 61,256)		Patients with CKD (*n* = 15,314)		Controls (*n* = 45,942)	
	*n*, %		*n*, %		*n*, %	
Yes	770	1.26%	591	3.86%	179	0.39%
No	60,494	98.74%	14,723	96.14%	45,771	99.61%
OR (95% CI)			10.249 (8.662~12.126)		1.000	

Notes: <0.001; OR = odds ratio.

**Table 3 jcm-11-07524-t003:** Covariate-adjusted odds of prior tinnitus (OR and 95% confidence interval, CIs) among the CKD group vs. controls (*n* = 61,264).

Variable	Odds of Having CKD		
	Adjusted OR	95% CI	*p* Value
Prior Tinnitus	10.970	9.255~13.004	<0.001
Age	0.980	0.978~0.982	<0.001
Sex	1.057	1.016~1.099	0.006
Monthly income			
<TWD 15,841 (reference group)	1.0	-	
TWD 15,841~25,000	0.826	0.786~0.867	<0.001
≥TWD 25,001	0.861	0.821~0.904	<0.001
Geographic region			
Northern (reference group)	1.0	-	
Central	0.920	0.873~0.969	0.002
Southern	0.967	0.922~1.015	0.179
Eastern	1.057	0.936~1.193	0.369
Urbanization level			
1 (reference group)	1.0	-	
2	0.964	0.916~1.014	0.156
3	0.986	0.929~1.047	0.641
4	0.949	0.886~1.015	0.129
5	0.948	0.885~1.016	0.129
Hyperlipidemia	1.398	1.327~1.472	<0.001
Diabetes	1.377	1.299~1.459	<0.001
Hypertension	1.187	1.133~1.243	<0.001
Coronary heart disease	1.087	0.998~1.184	0.055
Obesity	1.068	0.793~1.439	0.664
Urinary tract obstruction	-	-	
Autoimmune disease	0.883	0.764~1.021	0.093

## Data Availability

Data from the National Health Insurance Research Database, now managed by the Health and Welfare Data Science Center (HWDC), can be obtained by interested researchers through a formal application process addressed to the HWDC, Department of Statistics, Ministry of Health and Welfare, Taiwan (https://dep.mohw.gov.tw/DOS/lp-2506-113.html, accessed on 2 January 2022).

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
