# Peer review of "Association of Chronic Kidney Disease with Prior Tinnitus: A Case–Control Study"

_jcm, 2022, doi:10.3390/jcm11247524_

Round 1

Reviewer 1 Report

The study is a case control study of 15,314 patients with Chronic Kidney Disease (CKD) and 45,942 matched controls. By using the Taiwan National Health Insurance Research Database (NHIRD) the researchers get an unbiased and representative sample of the population. The study shows that patients diagnosed with tinnitus are 10.97 times more likely to belong to the CKD group when correcting for other health problems.

The study is very well written, the language is clear and concise. The paper is well structured and easy to understand. In summary, it was a pleasure to read.

Main points

There are no major problems in the study that should prevent publication, but there are some points that would be interesting if they could be included.

1. How long does it take from Tinnitus diagnosis to CKD diagnosis? Is there data for this? It is clear, that this is a difficult question to answer but if we want to use tinnitus as a marker of CKD this would be very relevant. Maybe it is possible to get an average time between Tinnitus diagnosis and CKD diagnosis. I do not know how the dataset is organized so I have no way of knowing if this is possible. Even a vague estimate would be nice.

2. A point that is not mentioned in the paper that might be important to stress for the reader that is not a hearing specialist is that this paper defines Tinnitus as patients with a diagnosis of Tinnitus and at least two claims showing Tinnitus within 3 years. This is most likely a quite severe tinnitus. Just asking patients if they have Tinnitus or experiences a ringing sound would not be the same thing. I mostly see this as a problem if anybody wants to use questions about tinnitus as a screening question. In several countries it has been found that more than half the population experiences tinnitus while the number of patients with a tinnitus diagnosis is only around 3% (doi:10.1001/jamaneurol.2022.2189).

3. Cases and controls seem to well matched, but it is a bit worrying that they are significantly worse when looking at: Diabetes, Hypertension, Coronary heart disease, Hyperlipidemia and Obesity. If it had been only one or two of them, I would not have seen that as a problem because the differences are quite small individually. But in aggregation the controls seem healthier. Since the odds ration still is ~11 after correcting for these factors this can clearly not be the cause of the difference in tinnitus. (Actually, the authors discuss this very well and I only left this comment in since it is something many readers will notice)

Minor points
This is minor details like spelling mistakes or minor errors.

Table 3 The thin lines in the figure are confusing. I expect them to separate sections, but one is above “Geographic region”, another is below “Urbanization level” and the last is between “central” and “southern”. What are they supposed to mark?

Line 29 “adjsuted analysis” is probably a typo

Line 131 remove "was used" or reword sentence

Line 226 “Wre” is probably a typo

Author Response

The study is a case control study of 15,314 patients with Chronic Kidney Disease (CKD) and 45,942 matched controls. By using the Taiwan National Health Insurance Research Database (NHIRD) the researchers get an unbiased and representative sample of the population. The study shows that patients diagnosed with Tinnitus are 10.97 times more likely to belong to the CKD group when correcting for other health problems.

The study is very well written, the language is clear and concise. The paper is well structured and easy to understand. In summary, it was a pleasure to read.

Response: Thank you very much.

Main points

There are no major problems in the study that should prevent publication, but there are some points that would be interesting if they could be included.

  1. How long does it take from Tinnitus diagnosis to CKD diagnosis? Is there data for this? It is clear, that this is a difficult question to answer but if we want to use Tinnitus as a marker of CKD this would be very relevant. Maybe it is possible to get an average time between Tinnitus diagnosis and CKD diagnosis. I do not know how the dataset is organized so I have no way of knowing if this is possible. Even a vague estimate would be nice.

Response: Thanks for your valuable suggestions. After analyzing the data, we found that of 591 patients with CKD, the duration between the first diagnosis of CKD and the first diagnosis of tinnitus was 803 ± 547 days. We have added the relevant statements into the Results (lines 155~157).

  1. A point that is not mentioned in the paper that might be important to stress for the reader that is not a hearing specialist is that this paper defines Tinnitus as patients with a diagnosis of Tinnitus and at least two claims showing Tinnitus within 3 years. This is most likely a quite severe tinnitus. Just asking patients if they have Tinnitus or experiences a ringing sound would not be the same thing. I mostly see this as a problem if anybody wants to use questions about Tinnitus as a screening question. In several countries it has been found that more than half the population experiences Tinnitus while the number of patients with a tinnitus diagnosis is only around 3% (doi:10.1001/jamaneurol.2022.2189).

Response: Thank you for your comments. We totally agree that as tinnitus is addressed with higher threshold, it would be less useful to serve as a screening indicator. We have stressed the above mentioned points in the manuscript as follows: “Another point that is important to be stressed is that in our study, we define Tinnitus as patients with a diagnosis of Tinnitus and at least two claims showing Tinnitus within 3 years and thus might be regarded as quite severe Tinnitus. This defined Tinnitus might be quite different from patients who ever experienced a ringing sound in their ears and could be a problem if anybody wants to use questions about Tinnitus as a screening question.”(lines 267~275)

  1. Cases and controls seem to well matched, but it is a bit worrying that they are significantly worse when looking at: Diabetes, Hypertension, Coronary heart disease, Hyperlipidemia and Obesity. If it had been only one or two of them, I would not have seen that as a problem because the differences are quite small individually. But in aggregation the controls seem healthier. Since the odds ration still is ~11 after correcting for these factors this can clearly not be the cause of the difference in Tinnitus. (Actually, the authors discuss this very well and I only left this comment in since it is something many readers will notice)

Response: Thanks for your comments. We have the following statements in Results: “Although there were significant differences, all effect sizes are small (≤ 0.2), considered too small to be of practical significance.”(lines 147~149)

Minor points

This is minor details like spelling mistakes or minor errors.

Table 3 The thin lines in the figure are confusing. I expect them to separate sections, but one is above “Geographic region”, another is below “Urbanization level” and the last is between “central” and “southern”. What are they supposed to mark?

Response: We have deleted the thin lines in Table 3.

Line 29 “adjsuted analysis” is probably a typo

Response: we have corrected the typo.

Line 131 remove "was used" or reword sentence

Response: We have removed them accordingly.

Line 226 “Wre” is probably a typo

Response: we have corrected the typo.

Reviewer 2 Report

1. The introduction is missing some background about tinnitus; prevalence and major etiologies. The pathophysiology of tinnitus is not well known, although often connected with hearing impairment and noise exposure,  tinnitus does not always derive from the inner ear. 

2. The study design based on health insurance claims that may explain the low prevalence of tinnitus in both cases and controls. Population based studies show a prevalence of 10-20%. One may presume that subjects seeking medical service have a tinnitus of more severe degree. 

3. The results are not surprising since the comorbidity between kidney failure and hearing impairment is well documented, and hearing impairment often include tinnitus, more so in high frequency loss. It is, however, remarkable that there was an increased claim for tinnitus prior to CKD.

4. Despite these interesting results, the conclusion that all patients with tinnitus should be investigated for CKD need better support, given the high prevalence of tinnitus in the population. The majority of tinnitus is not caused by CKD. Possibly, more data about the degree of tinnitus and hearing loss in these subjects could provide a prognostic tool. 

Author Response

Comments and Suggestions for Authors

  1. The introduction is missing some background about Tinnitus; prevalence and major etiologies. The pathophysiology of Tinnitus is not well known, although often connected with hearing impairment and noise exposure, Tinnitus does not always derive from the inner ear.

Response: We have revised the introduction to include info regarding the background about Tinnitus as follows: “While multiple pathophysiologies were identified, including inner ear pathology, auditory nerve synchronization, central nervous system anomalies, and limbic and autonomous nervous system problems, the Tinnitus, with a prevalence range from 7.1% to 14.6%, remained poorly understood for its underlying mechanisms responsible for the development of this abnormal sensory state (15).”(lines 64~68)

  1. The study design based on health insurance claims that may explain the low prevalence of Tinnitus in both cases and controls. Population based studies show a prevalence of 10-20%. One may presume that subjects seeking medical service have a tinnitus of more severe degree.

Response: Thank you for the comments. Similar to the questions raised by another reviewer, we have added the statements in the manuscript as follows: “Another point that is important to be stressed is that in our study, we define Tinnitus as patients with a diagnosis of Tinnitus and at least two claims showing Tinnitus within 3 years and thus might be regarded as quite severe Tinnitus. This defined Tinnitus might be quite different from patients who ever experienced a ringing sound in their ears and could be a problem if anybody wants to use questions about Tinnitus as a screening question.” (lines 267~275)

  1. The results are not surprising since the comorbidity between kidney failure and hearing impairment is well documented, and hearing impairment often include Tinnitus, more so in high frequency loss. It is, however, remarkable that there was an increased claim for Tinnitus prior to CKD.

Response: Thank you for the comments and we agree with you that a claim for Tinnitus prior to CKD might be of greater clinical value compare to identifying possible comorbidities after the development of CKD.

  1. Despite these interesting results, the conclusion that all patients with Tinnitus should be investigated for CKD need better support, given the high prevalence of Tinnitus in the population. The majority of Tinnitus is not caused by CKD. Possibly, more data about the degree of Tinnitus and hearing loss in these subjects could provide a prognostic tool.

Response: Thank you for the comments. We have added the statements in the manuscript as follows: “Without higher selection criteria, a high prevalence of Tinnitus is expected in the population, and as CKD does not cause the majority of Tinnitus, to increase the applicability, more data about the degree of Tinnitus and hearing loss in these subjects might be necessary for the phenomenon to serve as a proper prognostic tool.”(lines 272~275)

Reviewer 3 Report

I guess that despite your honest effort this manuscript is absolutely out of any realistic conclusion. You admitted a lot of bias in the discussion section indeed. First of all, there are no data about tinnitus, its possible origin, risk factors, otologic diseases, otoscopic evaluation, etc. There is no reliable reason that a patient affected by tinnitus may develop a CKD without any possible and logical link. May be the contrary..... 

Author Response

I guess that despite your honest effort this manuscript is absolutely out of any realistic conclusion. You admitted a lot of bias in the discussion section indeed. First of all, there are no data about Tinnitus, its possible origin, risk factors, otologic diseases, otoscopic evaluation, etc. There is no reliable reason that a patient affected by Tinnitus may develop a CKD without any possible and logical link. May be the contrary.....

Response: We regret that reviewer 3 has raised heavy criticisms on this work. However, we would like to address here again that this study is observational with many natural limitations. While there is one only truth regarding the relationship between Tinnitus and CKD, this work serves as only one way to observe the truth and by no means that we intended to overstate that our result is the only truth. By disclosing these biases in the manuscript, readers should have a clear viewpoint in reading this work. Further studies are needed and should be prospective in nature, with the inclusion of data about Tinnitus, its possible origin, risk factors, otologic diseases, and otoscopic evaluation in the pursuit of the truth.

Round 2

Reviewer 3 Report

I am sorry to not satisfy Authors' expectations but despite a few inclusions I did not find relevant variations of the manuscript. In particular I strongly refuse to accept what also in the abstract is written "our study shows that persons with tinnitus are at higher risk of a subsequent diagnosis of CKD".   How could I imagine to advise one of my patients affected by tinnitus that in the future he/she may be suffer from CKD! Or that his/her probabilities of that disease could be higher than if he/she had not a symton calle tinnitus. It is easy and simple to remind that tinnitus may be found in patients suffering from Meniere's Disease, presbyacusis, otospongiosis, acoustic acute and chronic trauma, vestibular schwannoma, etc, etc, all kind of pathologies where a microvascular role is hardly to be present, at least as main or unique cause. I would eventually appreciate the study if tinnitus were clearly linked to a possibly vascular origin. In this case the association with CKD could be reliable.

Author Response

Comments and Suggestions for Authors

I am sorry to not satisfy Authors' expectations but despite a few inclusions I did not find relevant variations of the manuscript. In particular I strongly refuse to accept what also in the abstract is written "our study shows that persons with tinnitus are at higher risk of a subsequent diagnosis of CKD".  

Response: We agee that the statements in the abstract should be soften as is revised as follows:

“Our study shows that persons with Tinnitus are associated with at higher risk of a subsequent diagnosis of CKD.”

“Nevertheless, the high odds of CKD observed in our study suggest the need for considering exploratory attempts including surveillance for CKD and lifestyle precautionary measures among patients diagnosed with Tinnitus, and among persons with existing CKD, measures to halt disease progression.”

“In conclusion, our study finds that patients with prior Tinnitus are associated with a higher risk of a subsequent diagnosis of CKD.”

Please allow us to emphasize again here that the purpose of this study is only to describe a possible relationship between diseases to inspire future who my be interested to go deeper, but not to encourage physicians to change their clinical practices

How could I imagine to advise one of my patients affected by tinnitus that in the future he/she may be suffer from CKD! Or that his/her probabilities of that disease could be higher than if he/she had not a symton calle tinnitus.

Response: As mentioned above, at this stage, no clinical practice adjustments should be made according to these findings. However, we believe that it is not a bad thing for physicians to know our findings, just like many study results, might fade away if no further supporting evidences were reveals. Or else, might be of value if further evidences with different study designs were presented in the future.

It is easy and simple to remind that tinnitus may be found in patients suffering from Meniere's Disease, presbyacusis, otospongiosis, acoustic acute and chronic trauma, vestibular schwannoma, etc, etc, all kind of pathologies where a microvascular role is hardly to be present, at least as main or unique cause. I would eventually appreciate the study if tinnitus were clearly linked to a possibly vascular origin. In this case the association with CKD could be reliable.

Response: Thank you for the comments. However, this database observational study is not aimed for mechanism explorations by nature, and perhaps will do a better job in rasing discussions such as microvascular pathologies as you have mentioned, and in which should be considered as the aim of this study.